# Evaluation of Aluminium Hydroxide Nanoparticles as an Efficient Adjuvant to Potentiate the Immune Response against *Clostridium botulinum* Serotypes C and D Toxoid Vaccines

**DOI:** 10.3390/vaccines11091473

**Published:** 2023-09-10

**Authors:** Ziphezinhle Mbhele, Lungile Thwala, Thandeka Khoza, Faranani Ramagoma

**Affiliations:** 1Onderstepoort Biological Products, 100 Old Soutpan Road, Onderstepoort, Pretoria 0110, South Africa; ziphezinhle@obpvaccines.co.za (Z.M.); lthwala@csir.co.za (L.T.); 2Discipline of Biochemistry, School of Life Sciences, University of KwaZulu Natal, Pietermaritzburg Campus, Private Bag X01, Scottsville 3209, South Africa; khozat1@ukzn.ac.za; 3Council for Scientific and Industrial Research, National Laser Centre, Pretoria 0001, South Africa

**Keywords:** botulism, toxoids, vaccines, alhydrogel, nanoalum

## Abstract

*Clostridium botulinum* serotypes C and D cause botulism in livestock, a neuroparalytic disease that results in substantial economic losses. Vaccination with aluminium-based toxoid vaccines is widely used to control the spread of botulism. Aluminium-based adjuvants are preferred owing to their apparent stimulation of the immune responses to toxoid vaccines when compared to other adjuvants. The aim of our study was to evaluate aluminium hydroxide nanoparticles as a potential substitute for alhydrogel in the botulism bivalent vaccine. Botulism vaccines were formulated with either alhydrogel or nanoalum and comparative efficacy between the two formulations was conducted by evaluating the immune response in vaccinated guinea pigs. A significant increase in immunological parameters was observed, with the antibody titres higher in the serum of guinea pigs (20 IU/mL of anti-BoNT C/D) injected with nanoalum-containing vaccine than guinea pigs inoculated with the standard alhydrogel-containing vaccine (8.7 IU/mL and 10 IU/mL of anti-BoNT C and anti-BoNT D, respectively). Additionally, the nanoalum-containing vaccine demonstrated potency in a multivalent vaccine (20 IU/mL of anti-BoNT C/D), while the standard alhydrogel-containing vaccine showed a decline in anti-BoNT C (5 IU/mL) antibody titres.

## 1. Introduction

Botulism is a severe neuroparalytic disease caused by *Clostridium botulinum* toxins [1]. *C. botulinum* bacteria are divided into six serotypes (A–F) according to their genetic, physiological, and metabolic characteristics. Serotypes A, B, E, and, to a lesser extent, F, are responsible for human botulism, while animal botulism arise from infection with serotypes C and D, whereas *C. botulinum* C/D and D/C neurotoxin mosaics are mainly associated with avian botulism [2]. In livestock, botulism is mainly acquired through the ingestion of the toxins from contaminated water or silage. The disease may also be contracted through wound and intestinal toxic infections [3]. Upon infection, the botulism neurotoxins are absorbed into the bloodstream and carried to the peripheral cholinergic nerve terminal where they block the release of acetylcholine, resulting in the flaccid paralysis of the muscles [4]. The disease is characterised by progressive paralysis or onset death in severe cases.

Protection against botulinum intoxication relies on the presence of specific neutralizing antibodies during exposure [5]. In acute cases, antibody therapy is effective, but not always accessible due to high costs [6]. Consequently, vaccination is the most effective control measure for botulism. The most widely used vaccines are bivalent (*C. botulinum* C and D) vaccines that are formulated with formaldehyde-inactivated toxins derived from the fermentation of *C. botulinum* [7]. Outbreaks in vaccinated cattle have been a source of concern about the quality and efficacy of the existing vaccines [8] Recently, recombinant protein-based Botulinum vaccines have been evaluated as a potential alternative to the chemically treated toxoids. Recombinant vaccines have been shown to present low toxicity and good immunogenicity among other advantages [9,10]; however, these have not reached the market. The downstream processes used in the purification of recombinant antigens require expensive purification procedures, making them unattractive from an industrial perspective [7]. Therefore, innovative methods are still being explored to improve the efficacy of native Botulinum toxoids for vaccine production.

Current vaccines against botulism are formulated with aluminium hydroxide as an adjuvant [1]. Despite the controversy around its mechanism of action, aluminium hydroxide is the most well-established adjuvant and is often the first choice for vaccine development [11]. Recently, researchers have discovered a direct link between the size of the adjuvant material and the immune response. Adjuvants at the nanoscale have been shown to induce a robust and long-lasting immunity when compared to their microparticle counterparts [11,12]. In that context, prospects of aluminium hydroxide nanoparticles are emerging as potential vaccine adjuvants. Conventional aluminium hydroxide particles form aggregates of 1 to 10 µm size. These particles predominantly recruit antigen presenting cells (APCs) to the site of injection for the uptake of antigens and delivers them to the lymph node to generate an immune response. Aluminium hydroxide nanoparticles, on the other hand, range from 80 to 600 nm, and are therefore able to traffic to the lymph nodes where there is a dense population of APCs, improving the delivery of antigens to the APCs and the overall immune response [12,13]. This makes nanoalum an attractive alternative to conventional aluminium hydroxide, since antigen specific T cell activation is restricted within the lymph nodes [14].

The aim of this study was to evaluate the potency of aluminium hydroxide nanoparticles (nanoalum) as a potential substitute for conventional alhydrogel in a multicomponent vaccine. Alhydrogel and nanoalum were each formulated with BoNT C and BoNT D toxoids to form a bivalent botulism vaccine and tested in guinea pigs. The two adjuvants were further tested in a multivalent vaccine, containing the Botulinum toxoids and two additional bacterial antigens.

## 2. Materials and Methods

### 2.1. Preparation and Characterization of Nanoalum

Nanoalum was prepared from alhydrogel (Onderstepoort Biological Products (OBP), Pretoria, South Africa) used in commercial vaccines. The alhydrogel was mixed with 50% (*w*/*v*) of 5 kDa polyacrylic acid (PAA) (Sigma, Ronkonkoma, NY, USA) to a final concentration of 2.7% (*v*/*v*) [15]. The mixture was homogenised at 10,000 rpm for 10 min at room temperature (RT). The average size and polydispersity index (PDI) of the nanoalum and alhydrogel were determined via dynamic light scattering (DLS) and the zeta (ζ) potential was calculated from the electrophoretic mobility values determined using Laser Doppler Anemometry (LDA), BoNT, measured using a Zetasizer^®^ Nano-ZS, ZEN 3600, Malvern instruments, (Worcestershire, UK) equipped with a red laser light beam (λ = 632.8 nm). For the particle size and PDI measurements, the nanoalum and alhydrogel samples were diluted 50× in milliQ water (18.2 MΩ-cm), and for the *ζ*-potential they were diluted 50× in 1 mM of KCl. The shape and properties of the alhydrogel and nanoalum were analysed using a transmission electron microscope (TEM, Joel 2010, 80 kV, Philips, Amsterdam, The Netherlands). The samples were deposited on a copper grid and allowed to dry before viewing under the TEM. The viscosity and conductivity of the adjuvants were measured at RT using a viscometer (AMTEK Brookfield, Middleborough, MA, USA) at 5 rpm and 100 rpm for the alhydrogel and nanoalum, respectively.

### 2.2. Adsorption Capacity of Alhydrogel and Nanoalum

The antigen adsorption capacity of alhydrogel and nanoalum was analysed via SDS-PAGE using formaldehyde-inactivated *C. botulinum* Type D toxoid (OBP, SA) as a model antigen. The BoNT D toxoid vaccine was formulated with (10% *v*/*v*) either alhydrogel or nanoalum and stirred for 24 h at RT. The formulation was sampled at 0 h and 24 h; the samples were centrifuged at 13,000× *g*, 10 min, 4 °C. The supernatant was analysed on a 7.5% non-reducing SDS-PAGE according to the method described by Laemmli, 1970 [16]. The inactivated BoNT D toxoid was included as a reference (initial sample), and the same result would be expected for BoNT C.

### 2.3. Adjuvant Safety and Vaccine Formulation

Nanoalum and alhydrogel were mixed with saline (OBP) to a final concentration of 20% (*v*/*v*). The adjuvant mixtures were administered SC in guinea pigs at a 2 mL dose on day 0 and day 28 to mimic the vaccinations. The animals were monitored for up to 14 days after the second dose for any local reactions [10].

Bivalent vaccines *(C. botulinum* type C and D toxoids) and multivalent vaccines (containing BoNT C and D toxoids, and two additional antigens derived from *C. perfringens* and *B. anthracis*) were formulated to the minimum lethal dose (MLD_50_) of ×10^4^ and ×10^5^, respectively, as described by Schantz and Kautter [17], with 10% (*v*/*v*) of either the alhydrogel or nanoalum in a final volume of 200 mL. The vaccines were stirred at 100 rpm for 24 h at 4 °C. The formulated vaccines were tested for sterility by inoculating 20 mL of the thioglycolate broth medium and soya broth with 0.5 mL of the vaccine sample, and they were incubated for 14 days at 37 °C under aerobic and anaerobic conditions [10].

### 2.4. Immunisation of Guinea Pigs

Immunizations were carried out as described by Gil et al. (2013) [18] with minor modifications. Duncan Hartley female guinea pigs aged 3–4 months, weighing 200 to 310 g, were acclimatized for three days prior to immunization. The vaccine groups were divided as follows: Group A—bivalent alhydrogel, Group B—bivalent nanoalum, Group C—multivalent alhydrogel, Group D—multivalent nanoalum, and Group E—placebo, and each group was assigned eight guinea pigs. A volume of 2 mL of the vaccine, including the control, was administered subcutaneously (SC) on day 0 and a booster was administered SC on day 28. On day 35, blood samples were collected from the guinea pigs via cardiac puncture and centrifuged, 3000× *g*, 10 min, to obtain sera [18]. The sera from each group were pooled and stored at −20 °C until further use in toxin neutralization studies.

### 2.5. Evaluation of Neutralising Antibodies

Toxin neutralising antibodies were evaluated in a toxin neutralization assay in mice as described by Rosen et. al., 2016 [19]. Briefly, the sera obtained in Section 2.4 was used to prepare serial 1.2-fold dilutions of each antitoxin preparation. Standard antitoxin preparations were concurrently diluted to concentrations of 0.08, 0.10, 0.12, and 0.14 International Units per mL (IU/mL). All antitoxin dilutions were then incubated for 30 min at 37 °C with a toxin test dose of 10^6^ MLD_50_ BoNT/C and 10^5^ MLD_50_ BoNT/D toxins. The ability of the anti-Botulinum antitoxins to neutralise the toxins was studied using a mouse assay to evaluate their potency via intravenous challenge. Each mixture was injected into 8-week-old CD-1^®^IGS female mice weighing between 18 and 22 g at 0.2 mL per mouse with 4 mice assigned to each group. Survival was monitored for three days. Antitoxin potency was calculated based on the lowest dilution of antitoxin that failed to protect the animals, compared to that of the standard antitoxin. The animals were monitored for three days, and the number of live and dead mice were recorded for each serum dilution (Figure 1) [18]. Titrations were also performed with standard anti-toxins C and D as controls (1st British standard 01/508 Botulinum type C equine antitoxin, 01/510 Botulinum D equine antitoxin, NIBSC, Potters Bar, UK).

### 2.6. Statistical Analysis

R Console version 3.2.1 (analytical software) was used to determine the significant differences among the vaccine groups and the adjuvant groups, using a *t*-test on the mean values [20].

### 2.7. Ethics Statement

Ethical clearance for the animal procedures was obtained from the Onderstepoort Biological Products (OBP) Animal Ethics Committee (South African Veterinary Council Facility Registration Number: FR1514054) and the Department of Agriculture, Land Reform and Rural Development under Section 20 of the Animal Diseases Act (Act 35 of 1984), Protocol reference number:12/11/11(b)/1914(HP).

## 3. Results

### 3.1. Preparation and Characterisation of Nanoalum

The morphology of the adjuvants was analysed under a transmission electron microscope (Figure 2). The alhydrogel displayed filamentous particles which formed aggregates that were dispersed via homogenization (Figure 2A,B). The addition of PAA prior to homogenisation resulted in the formation of nanoalum, which displayed plate-like particles (Figure 2C). The properties and particle sizes of the adjuvants were analysed using dynamic light scattering (DLS). The untreated alhydrogel was 3400 nm with a PDI of 1.0 (Figure 3A) and a *ζ* potential of +10 mV (Figure 3B). Following homogenisation, the alhydrogel particle size was reduced to an average of 620 nm (Figure 3A) and the nanoalum had an average diameter of 265 nm, and the addition of PAA shifted the *ζ*-potential to −53 mV (Figure 3B). The viscosity of the alhydrogel was 471 cP with a conductivity of 164.1 µS/cm, and these were reduced to 2.94 cP (water-like consistency) and 5.60 µS, respectively, in the nanoalum samples (Figure 3C). Furthermore, the nanoalum particles maintained a stable size conformation over six months of storage at 4 °C, whereas the alhydrogel reaggregated to 1555 nm (Figure 3A). The untreated alhydrogel particles’ sizes could not be determined via DLS after 6 months.

### 3.2. Adsorption Capacity of Alhydrogel and Nanoalum

The adsorption capacity of alhydrogel and nanoalum was studied using the BoNT D toxoid, which appears as a wide band which is >250 kDa (after inactivation with formaldehyde), with a pI of 5.4. The toxoid was formulated in a total volume of 2 mL with 10% (*w*/*v*) of either alhydrogel or nanoalum, and allowed to incubate for 24 h. The formulation was sampled at 0 h and 24 h and clarified via centrifugation to obtain a supernatant. The resultant supernatant was analysed on a 7.5% non-reducing SDS-PAGE gel (Figure 4) for the presence of any unbound antigens. Due to the difference in charges, the toxoid was adsorbed to the alhydrogel immediately following formulation, while the nanoalum showed a low adsorption over the 24 h incubation period as indicated by the presence of the toxoid in the supernatant. This low adsorption was expected as the addition of PAA shifted the *ζ*-potential of the adjuvant to a negative charge.

### 3.3. Evaluation of the Potency of Alhydrogel and Nanoalum Vaccine Formulation

#### 3.3.1. Safety and Protective Efficacy of Alhydrogel- and Nanoalum-Containing Vaccines

The animals were inoculated 28 days apart with 20% (*v*/*v*) of nanoalum and alhydrogel and monitored over 42 days. No reactions were observed in any of the animals; thus, nanoalum was reported as safe for use in guinea pigs. Neither fungal nor bacterial contaminates were detected in any of the vaccine and control samples.

The potency of the alhydrogel formulation was compared with that of the nanoalum formulation. The botulism bivalent vaccine contained BoNT C and D antigens. The multivalent vaccine contained BoNT C and BoNT D antigens, and two other bacterial antigens. The overall protective efficacy of the alhydrogel- and nanoalum-containing vaccines was informed by the total number of mice that survived in the toxin neutralization assay. In the bivalent formulations, the nanoalum vaccine demonstrated 100% protection against a challenge with BoNT toxins, while the maximum protection achieved with the alhydrogel vaccine was only 60% against the BoNT C and BoNT D challenge (Figure 5). In the multivalent vaccine study, the alhydrogel-containing vaccine achieved 40% and 55% protection against the BoNT C and BoNT D challenge, respectively, while 100% of the mice survived in the nanoalum vaccine group.

#### 3.3.2. Quantification of Toxin-Neutralizing Antibodies

The neutralizing antibody titres were quantified using toxin neutralization assay mice. The bivalent alhydrogel formulation induced 8.7 IU/mL of neutralizing antibodies against the BoNT C toxin, and 10 IU/mL against the BoNT D toxin. The bivalent nanoalum formulation was able to induce 20 IU/mL of neutralizing antibodies against the two toxins (Figure 6A,B). In the multivalent alhydrogel vaccine, anti-BoNT C showed a significant decrease (*p* < 0.0001) in antibody titres, from 8.7 IU/mL to 5 IU/mL, while anti-BoNT D titres were maintained at 10 IU/mL. Interestingly, the nanoalum formulation maintained 20 IU/mL antibody titres against the BoNT toxoids (Figure 6A,B). The BoNT alhydrogel and nanoalum vaccines were able to surpass the minimum required antibody levels, i.e., 5 IU/mL of anti-BoNT C and 2 IU/mL of anti-BoNT D [19]. There was a significant difference (*p* < 0.0001) between the antibody titres induced by the alhydrogel vaccines and nanoalum vaccines.

## 4. Discussion

Botulism continues to threaten livestock production; thus, routine immunizations are performed to control outbreaks. Recent reports of botulism outbreaks in vaccinated animals have raised concerns about the efficacy of vaccines against botulism [3,8,21]. There are several reports where commercial livestock Botulinum vaccines failed to elicit the required minimum antibody titres following their approval [22,23]. These observations highlight the need for alternative vaccine strategies to induce robust and durable protective immunity against botulism. Therefore, this study was aimed at evaluating nanoalum as a potential substitute for alhydrogel to improve the efficacy of Botulinum toxoid vaccines when administered as a bivalent vaccine or in combination with other antigens.

To date, Botulinum toxoid vaccines are predominantly adjuvanted with aluminium salts to aid the induction of a protective immune response [1]. The aluminium hydroxide particles disaggregate during mixing, thus allowing a uniform distribution of the adsorbed antigen in the vaccine [24]. This is a valuable characteristic, especially for vaccines which contain multiple antigens that are likely to compete for adsorption [24].

Resting alhydrogel readily reaggregates after mixing [25]; therefore, the size reported for alhydrogel after homogenisation may indicate a disaggregated system which continues to form aggregates over time. The size of the aggregates may vary, depending on the conditions of homogenisation; as a result, aluminium hydroxide particles are highly heterogenous and sometimes too big to characterise using dynamic light scattering [25]. Several studies have reported on the application of aluminium hydroxide nanoparticles to optimize the adjuvant activity in vaccines [15,26,27]. Unlike the conventional aluminium adjuvants, nanoalum particles are structurally well-defined and easier to characterise for quality control purposes. In the present study, the addition of PAA contributed to the uniformity and maintained the stability of the particles at 4 °C over a 6-month period. Similar results have been reported, where PAA contributed to the formation and stabilization of the nanoparticles at different storage conditions [15]. PAA-stabilized nanoalum has also demonstrated the ability to withstand multiple freeze–thaw cycles, making it an ideal adjuvant for a cold-chain environment. However, the stability of nanoalum vaccines at different temperatures must be carefully investigated as alum-containing vaccines are known to lose potency upon freeze–thawing [25].

The potency of alhydrogel is generally attributed to its interaction with antigens [28]. Acidic antigens adsorb onto alhydrogel based on electrostatic charge. Adsorption may result in partial unfolding, exposing additional hydrophobic residues, which further strengthen the adsorption [24]. The antigens are then slowly released in the interstitial fluid for a prolonged exposure to the immune system and induce type 2 (Th2)-biased immune responses [29,30]. Nanoparticles, on the other hand, allow for a greater adsorption capacity due to the larger surface area-to-volume ratio [13]. In this study, PAA-stabilised nanoalum showed a weaker adsorption capacity due to the shift in electrostatic charge. However, the was a stronger immune response when compared to alhydrogel. These results agree with a series of studies that reported an inverse relationship between the strength of antigen adsorption and the resultant immune response [13,15,31,32]. It is the effective adsorption of the antigen in the interstitial fluid that improves the immunological response [32]. The small size of nanoparticles facilitates a more efficient accumulation of the vaccine in the lymph nodes and induces a more balanced (Th1/Th2), robust, and durable immune response [30]. The eluted antigens are taken up by the antigen-presenting cells via micropinocytosis while the bound antigen is internalised through phagocytosis. Phagocytosis was shown to be more efficient compared to micropinocytosis in antigen uptake and was enhanced when the adjuvant aggregates were smaller [33]. Furthermore, it is important to highlight the relationship between the low viscosity of the nanoalum and the induced immune response observed in this study. This is because high viscosity has been a major constraint in the widespread application of some potent vaccines due to the reactogenicity at the site of injection [34]. Even though alum-based adjuvants are the key benchmark for the evaluation of new adjuvants, they have also been associated with some adverse side effects [27]. A good adjuvant should demonstrate a balance between safety and efficacy; therefore, the water-like consistency of nanoalum can be expected to minimise local reactions at the injection site. Additionally, the nanoalum suspension also maintained homogeneity during incubation, while alhydrogel particles readily separated from the liquid phase, forming a heterogenous mixture (results not shown).

Toxoid vaccines must induce specific levels of antibody titres in immunized animals to certify their efficacy [35]. In the case of botulism, 5 IU/mL and 2 IU/mL are the minimum required neutralising antibody levels for BoNT C and BoNT D, respectively. The formulations evaluated in this study exceeded these requirements, with the exception of BoNT C in the alhydrogel-adjuvanted multivalent vaccine that induced the minimum required antitoxin level. Moreover, there was a significant decrease (*p* < 0.0001) in the anti-BoNT C titres from the bivalent vaccine when compared to the multivalent vaccine. This observation is common with vaccines containing two or more antigens due to the antigenic competition between the antigens [24]. This phenomenon has been reported for BoNT C and D toxoids vaccines; thus, these antigens must be carefully proportioned to ensure a sufficient immune response to each component [22]. Additionally, BoNT D is more immunogenic when compared to BoNT C, thus eliciting higher antibody titres, as reported in several studies [22,23,36]. Interestingly, this variation was not observed with the vaccines containing nanoalum, as the BoNT antigens achieved the same levels of toxin-neutralising antibody titres. Comparable anti-BoNT D titres have been achieved with a recombinant bivalent botulism vaccine, while the immunogenicity of BoNT C remained inferior [9]. Moreover, the potency demonstrated by nanoalum in the bivalent vaccine was maintained in the multivalent vaccine. Although experimental models do not always accurately mimic the immune responses in target animals [37], the nanoalum vaccines used in this study have the potential to meet or exceed the minimum required protective immunity in target animals. Nevertheless, further experiments in field animals are required to evaluate the extent to which the nanoalum-containing vaccines induce an immunological response.

In addition to a stronger IgG immune response, nanoalum has also demonstrated the ability to augment a Th1 immune response in ways that micron-sized aluminium hydroxide could not [15,38]. The findings reported here also highlight the improved antigen delivery offered by nanoalum. This study further reports that nanoalum could be an ideal alternative, where the antigen concentrations exceed the absorption capacity of the standard alum-based adjuvants, such as in multicomponent vaccines [24]. However, the potency of nanoalum still requires critical evaluation. A study conducted previously reported that the enhanced immune response in a nanoalum-adjuvanted vaccine was associated with overactivation of the innate immune response as well as local and systemic reactogenicity [39]. Nevertheless, the reactogenicity may be addressed via dose reductions. The success of nanoalum in the potentiation of an immune response offers an added advantage of dose reduction.

## 5. Conclusions

In this study, we converted conventional alhydrogel to nanoalum and evaluated it as an efficient adjuvant to improve the immune response to botulism bivalent vaccines as well as in a multicomponent vaccine. Nanoalum demonstrated superior adjuvating ability when compared to alhydrogel and can be used to improve the immune response to botulism vaccines. Additionally, the subjugation of antigenic competition demonstrated by nanoalum in a multivalent vaccine could open a new avenue for vaccine development. While experimental trials conducted in experimental animals provide some information on the nanoalum, these same results cannot be guaranteed in target animals. Therefore, further studies are required to evaluate the safety, potency, and duration of the immune response conferred by nanoalum vaccines in livestock.

## Figures and Tables

**Figure 1 vaccines-11-01473-f001:**
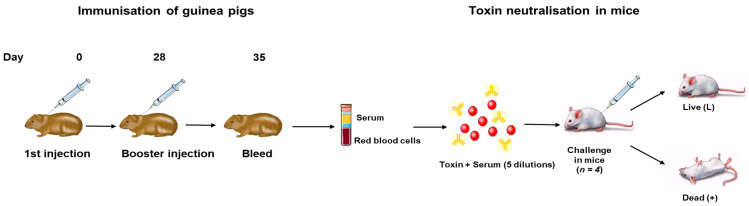
Schematic representation of immunisation of guinea pigs and toxin neutralization assay in mice. BoNT C and BoNT D were formulated as a bivalent vaccine as well as a multivalent vaccine with two additional antigens. The vaccine was administered subcutaneously on day 0 (1st injection) and day 28 (booster injection). On day 35, the guinea pigs were bled, and the blood samples were centrifuged to obtain sera, which was evaluated in a toxin neutralisation assay. Live mice indicate neutralisation of the toxin and death indicates the absence of toxin neutralisation.

**Figure 2 vaccines-11-01473-f002:**
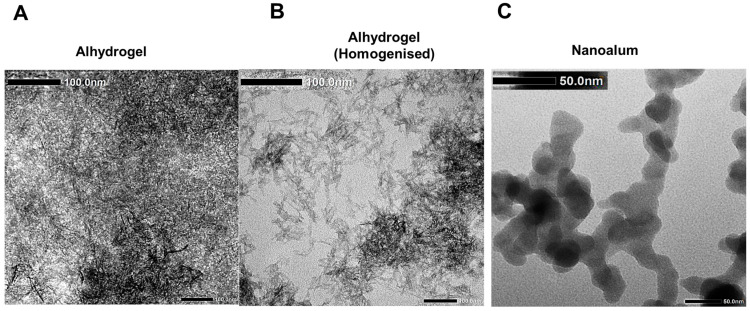
Characterization of alhydrogel and nanoalum particle morphology via transmission electron microscopy. (**A**) Untreated alhydrogel particles. (**B**) Alhydrogel after homogenization at 10,000 rpm. (**C**) PAA-stabilized nanoalum. The scale bar was 100 nm for alhydrogel and 50 nm for nanoalum.

**Figure 3 vaccines-11-01473-f003:**
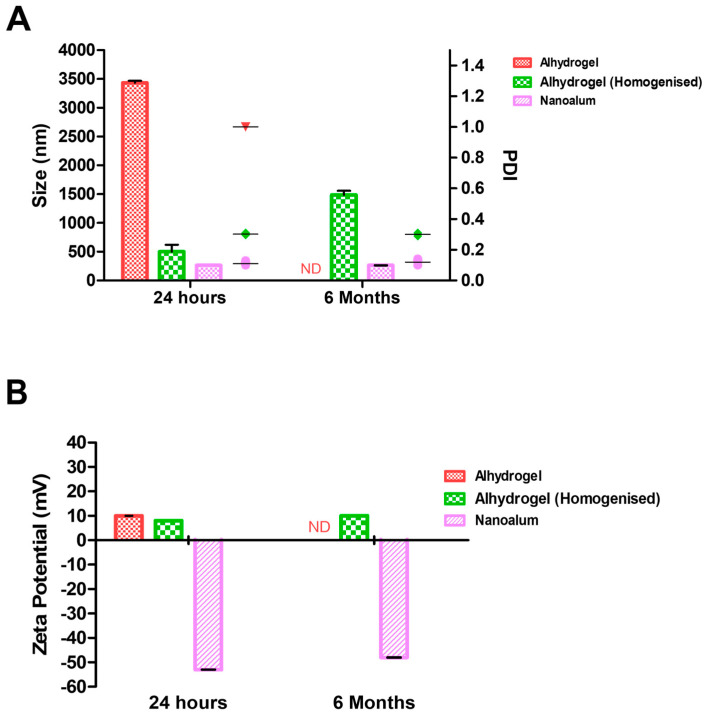
Physiochemical characterization of alhydrogel and nanoalum. (**A**) Particle sizes and polydispersity index (PDI), and (**B**) zeta potential analysis via dynamic light scattering on a zetasizer. (**C**) Viscosity and conductivity measured using a viscometer. ND: not determined.

**Figure 4 vaccines-11-01473-f004:**
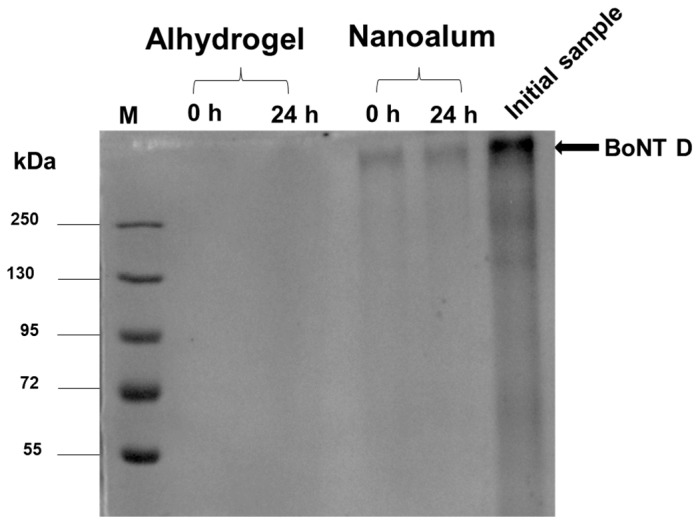
Adsorption capacity of alhydrogel and nanoalum analysed via SDS-PAGE. Alhydrogel and nanoalum were formulated with *C. botulinum* type D toxoid and incubated for 24 h. Samples were taken at 0 h and 24 h, centrifuged, and the resulting supernatants were analysed on a 7.5% non-reducing SDS-PAGE. M: Molecular weight marker. Initial sample: inactivated toxoid sample before formulation.

**Figure 5 vaccines-11-01473-f005:**
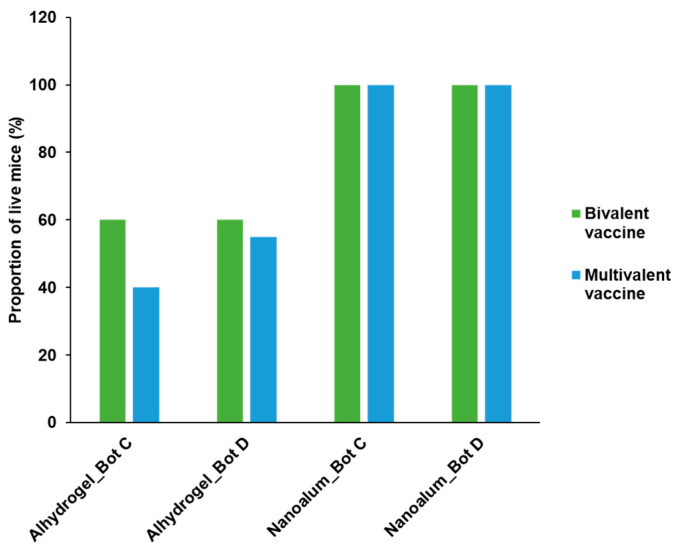
Protective efficacy of alhydrogel- and nanoalum-adjuvanted botulism vaccines in a toxin neutralization assay. The antisera from guinea pigs vaccinated with the different vaccine groups (alhydrogel- and nanoalum-adjuvanted vaccines) were evaluated in a mice challenge against MLD_50_ of BoNT C and D toxins. Bivalent vaccine: containing BoNT C and BoNT D, Multivalent vaccine: BoNT C, BoNT C, and two other bacterial antigens.

**Figure 6 vaccines-11-01473-f006:**
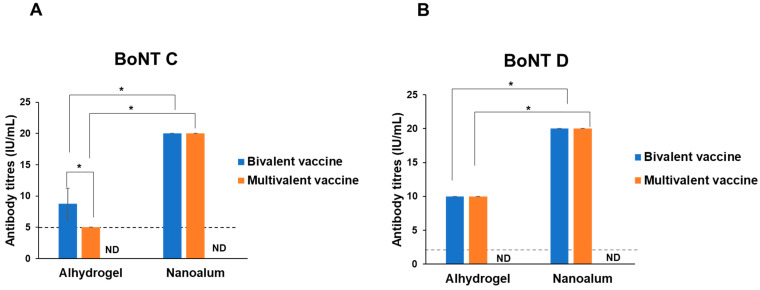
Neutralizing antibody titres against BoNT C and D toxins induced by alhydrogel- and nanoalum-adjuvanted vaccines. The guinea pigs were immunized with a Botulinum type C and D bivalent vaccine; seven days after booster immunization, the guinea pigs were bled, and the neutralizing antibodies obtained from the sera were quantified in a toxin neutralization assay. (**A**) Anti-BoNT C titres. (**B**) Anti-BoNT D titres. The dotted lines represent the minimum required titres. Bivalent vaccine: containing BoNT C and BoNT D; Multivalent vaccine: BoNT C, BoNT C, and two other bacterial antigens ND: not detected. * *p* < 0.0001.

## Data Availability

Not applicable.

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
