# Peer review of "Evaluation of Aluminium Hydroxide Nanoparticles as an Efficient Adjuvant to Potentiate the Immune Response against Clostridium botulinum Serotypes C and D Toxoid Vaccines"

_vaccines, 2023, doi:10.3390/vaccines11091473_

Round 1

Reviewer 1 Report

Authors compared aluminium hydroxide nanoparticles versus aluminium based adjuvant in the case of  immune response against Clostridium botulinum serotypes C and D. As indicated immune response to vaccine formulated with nanoparticle was increased in immunological parameters.

Although results are interesting but limited evaluations in methods (only neutralization test) and the low number of case and samples (pooled serum and only one blood at 35 day) as well as no toxicity test or different concentration of antigen cannot support a well scientific finding.

Author Response

Responses on the attached file. 

Reviewer 2 Report

Applications of nanotechnology has revolutionized medical science. Nano particles could be used in for many purposes including effective vaccine performance, drug deliver, combat AMR etc. Here we also see the application of nanotechnology. It’s a nice study where the authors tried to evaluate the efficacy of aluminium hydroxide nanoparticles as adjuvants to induce better immune response against Cl. botulinum serotypes C and D toxoid vaccines.

Comments:

Title: use ‘an’ between as and efficient…so it will  be “as an efficient’’’’

Abstract: introduction section  (line 13-17) is too long, please make it short

All the pictures in Fig 2 is of poor quality, please replace with high-resolution pictures

Provide a reference to support the use of method: 2.4. Immunisation of guinea pigs

Fig 3. What do you mean by initial sample, please explain in methodology

Line 152, how do you confirm the average size at 620 nm

 Discussion section is poor, first two para are just like the story of other people…should focus  and linked with findings of this study too… so rewrite DISCUSSION, please.

Please show the physical property of the nanoparticle in fig/graphical e.g., a plot  showing presentation of Particle sizes (open bar) and zeta potentials plot. This will further conform the nanopartcle formation.

Usually, people also do X-ray diffraction of aluminum hydroxide particles, could you please explain why it was not done here? 

Author Response

Responses on the attached file. 

Reviewer 3 Report

This is a clear, well-written manuscript with interesting positive results on an alternative adjuvant for types C & D toxoid animal vaccines.

Recommendations:

40                    Botulinum toxin (BoNT) actually BLOCKS, not inhibits, acetylcholine transfer

43-44              Citation needed for antibody use in animals

89                    MilliQ water should be reported as the quality, not how it is made

Section 2.2    The authors should comment on why they chose type D toxoid for this measurement and also if they would expect the same result with type C toxoid

107                 How was the MLD50 measured?

Section 2.4    What was the sex and weight of the guinea pigs used?  Why was such a wide age range used?

120                 Citation missing

Section 2.5    What was the sex and age of the mice used?  What was the breed?

128                 Citation missing

Figure 1 should be referenced in Sections 2.4 and 2.5

Section 3.2    This section examines the adsorption of both alhydrogel and nanoalum and so should be titled for both

169                 What effect, if any, would the centrifugation have on the adsorbed toxoid?  Would the centrifugation have removed the alhydrogel adsorbed toxoid, which would give the result obtained on the gel analysis?

None

Author Response

Responses on the attached file. 
